# The Minute Virus of Canines (MVC) Activates the RhoA/ROCK1/MLC2 Signal Transduction Pathway Resulting in the Dissociation of Tight Junctions and Facilitating Occludin-Mediated Viral Infection

**DOI:** 10.3390/microorganisms13030695

**Published:** 2025-03-20

**Authors:** Xiang Ren, Zhiping Hei, Kai Ji, Yan Yan, Chuchu Tian, Yin Wei, Yuning Sun

**Affiliations:** School of Basic Medical Science, Ningxia Medical University, Yinchuan 750004, China; 13895482903@163.com (X.R.); hax163guoker@163.com (Z.H.); ssjyhhxx@163.com (K.J.); eileen_yanyan@163.com (Y.Y.); tiancc07@163.com (C.T.); 13665240385@163.com (Y.W.)

**Keywords:** MVC, protein interaction, RhoA/ROCK1/MLC2 signaling pathway, virus entry, VP2 protein

## Abstract

The Minute Virus of Canines (MVC), belonging to the genus *Bocaparvovirus* within the family *Parvoviridae*, is associated with enteritis and embryonic infection in neonatal canines. Viral attachment to host cells is a critical step in infection, and viral protein 2 (VP2) as an important structural protein of MVC influences host selection and infection severity. Nevertheless, little is known about the interaction between VP2 protein and host cells. In this study, we identified that VP2 directly interacts with the kinase domain of RhoA-associated protein kinase 1 (ROCK1) by using mass spectrometry and immunoprecipitation approach and demonstrated that the RhoA/ROCK1/myosin light chain 2 (MLC2) signaling pathway was activated during the early stage of MVC infection in Walter Reed canine cell/3873D (WRD) cells. Further studies indicated that RhoA/ROCK1-mediated phosphorylation of MLC2 triggers the contraction of the actomyosin ring, disrupts tight junctions, and exposes the tight junction protein Occludin, which facilitates the interaction between VP2 and Occludin. Specific inhibitors of RhoA and ROCK1 restored the MVC-induced intracellular translocation of Occludin and the increase in cell membrane permeability. Moreover, the two inhibitors significantly reduced viral protein expression and genomic copy number. Collectively, our study provides the first evidence that there is a direct interaction between the structural protein VP2 of MVC and ROCK1, and that the tight junction protein Occludin can serve as a potential co-receptor for MVC infection, which may offer new targets for anti-MVC strategies.

## 1. Introduction

Minute Virus of Canines (MVC) belongs to the genus *Bocaparvovirus* in the *Parvoviridae* family [1]. The genus *Bocaparvovirus* includes members such as MVC, Bovine Parvovirus (BPV), and Human Bocavirus 1-4 (HBoV1-4), as well as several newly found species: Feline Bocavirus (FBoV), Animal Bocavirus (PBoV), Rat Bocavirus, and Mink Bocavirus [2,3,4,5]. MVC was first identified in canine fecal samples in 1967 and has since been detected in fecal or serum samples from domestic dogs of various ages worldwide [6,7,8]. As a significant pathogen impacting newborn puppies and canine fetuses, MVC may cross the placenta, causing embryonal mortality, spontaneous abortion, and congenital malformations in puppies [9,10], while also inducing severe gastroenteritis, myocarditis, and pneumonia in juvenile and immunocompromised hosts [11,12]. Recent evidence further implicates MVC in the pathogenesis of degenerative hepatitis in canines [13]. Given the worldwide adverse impact of MVC infection on dogs, further investigation into its pathogenic mechanisms and transmission characteristics will be useful for developing effective anti-MVC strategies.

Like other bocaviruses, the MVC genome (GenBank accession number: NC_075119.1) spans 5402 nucleotides flanked by characteristic palindromic hairpin structures (5′:183 nt; 3′:198 nt) and contains three open reading frames (ORFs) [14,15]. The two ORFs on the left and middle encode nonstructural proteins NS1 and NP1, respectively, which coordinate viral replication and transcriptional regulation [16,17], whereas the right ORF expresses overlapping capsid proteins VP1 and VP2 through alternative splicing. The MVC capsid, like with other parvoviruses, functions as a barrier of protection for the enclosed genome during the infection process, which begins with receptor attachment and cellular entry, involving the whole viral life cycle [18,19]. For host cell recognition the capsid attaches to specific receptor(s) on the cell surface. Although Blackburn et al. reported that BPV specifically binds to O-linked α2,3-linked sialic acids on GPA [20], the cell entry mechanisms mediated by the capsid proteins of other bocavirus species remain poorly characterized. The full length of the MVC genome exhibits 43% and 52.6% identity with the genomes of BPV and HBoV, respectively, sharing similarities in both genomic organization and pathogenic mechanisms [14]. Based on the fact that MVC infection of Walter Reed canine cell/3873D (WRD) allows for robust viral replication and the production of progeny viruses [21,22], the WRD cell line has become an ideal in vitro model for studying the pathogenesis of MVC. MVC, with its genetic tractability and reliable infectivity, is a suitable model for studying the interactions between the MVC capsid protein and the host, and for elucidating the infection mechanisms of bocaviruses.

It is well known that epithelial cells typically function as a barrier to protect against pathogen invasion through cell junctions. Tight junctions (TJs), dynamic multiprotein complexes regulating paracellular permeability, serve dual roles as physical barriers and potential viral entry portals [23,24,25]. The integrity of TJs relies on the synergistic interactions of Claudin, Occludin, and zonula occludens-1 (ZO-1) proteins. Meanwhile, their paracellular permeability is dynamically regulated through phosphorylation-dependent cytoskeletal remodeling, such as myosin light chain (MLC) activation, along with other post-translational modification mechanisms [26,27]. Notably, the RhoA/RhoA-associated protein kinase 1 (ROCK1)/MLC2 signaling pathway, a master regulator of actomyosin contractility, is exploited by multiple enteric viruses (e.g., porcine sapovirus, rotavirus, and PEDV) to dismantle TJs and facilitate transepithelial invasion [23,28,29]. ROCK1 is a serine/threonine kinase activated by RhoA-GTP, and increasing evidence implicates ROCK1 in diverse viral life cycles, including Hepatitis B virus (HBV) entry, human immunodeficiency virus (HIV) nuclear transport, and influenza virion release [30,31,32]. However, its role in bocavirus infections remains unexplored. In this study, we used the MVC capsid protein VP2 as bait and employed immunoprecipitation (IP) technology combined with mass spectrometry (MS) analysis to screen and identify host cell proteins interacting with MVC VP2. For the first time, we identified ROCK1 as an interacting partner of MVC VP2. Moreover, ROCK1 plays a significant role in regulating MVC entry.

## 2. Materials and Methods

### 2.1. Cells and Viruses

The MVC, WRD cell line and COS-1 cells were stored in our laboratory. The MVC and WRD cell lines were generously provided by Professor Jianming Qiu from the Department of Microbiology, University of Kansas Medical Center. Cells were cultured in Dulbecco’s Modified Eagle Medium (DMEM; Catalog No. FI101-01, TransGen, Beijing, China) supplemented with 10% fetal bovine serum (FBS; Catalog No. FS201-02, TransGen) and 1% penicillin–streptomycin (Catalog No. 15140122, Thermo Fisher Scientific, Waltham, MA, USA), under standard conditions (37 °C, 5% CO_2_, 95% humidity). WRD cells were exclusively used for viral titer determination and infectivity assays. Viral titers were quantified using a focus-forming assay (FFA) as described previously [33]. For infection experiments, WRD cells were inoculated with MVC at a multiplicity of infection (MOI) of 1.

### 2.2. Plasmid Construction

DNA fragments encoding three functional domains of ROCK1 (kinase domain, KD; Rho-binding domain, RBD; pleckstrin homology/cysteine-rich domain, PH-CRD; NCBI Reference Sequence: XM_005623022.4) and Occludin (NCBI Reference Sequence: U49221.1) were amplified by PCR. The oligonucleotide primers containing restriction enzyme sites are listed in Table 1. To construct the pCMV-HA-KD/-RBD/-PH-CRD and pCMV-HA-Occludin plasmids, the four PCR products were double-digested with *Sal*I/*Xho*I and cloned into the corresponding *Sal*I/*Xho*I sites of the pCMV-HA vector; the pGEX-4T-1-KD was constructed using the *Eco*RI/*Xho*I site of the pGEX-4T-1 vector. The ‘HA’ refers to the hemagglutinin tag. Both pCMV-HA and pGEX-4T-1 vectors were maintained in our laboratory. All plasmids were verified by restriction enzyme digestion analysis, and sequencing was confirmed by Sangon Biotech (Shanghai, China). The eukaryotic expression plasmid pXJ40-Flag-VP2 (encoding full-length VP2) and VP2-specific primers were generated as described previously [33].

### 2.3. Immunoprecipitation (IP) and Mass Spectrometry (MS) Analysis

WRD cells were cultivated in a 10 mm dish, infected with MVC for 48 h, lysed for 15 min on ice in RIPA buffer (Catalog No. 89900, Thermo Fisher Scientific) containing protease inhibitors, and then lysed at 14,000 g for 15 min at 4 °C to clarify. Protein A/G agarose beads (Catalog No. PR40025, Proteintech, Wuhan, China) were added to the supernatants after they had been treated with 5 μg of anti-VP2 antibodies or mouse IgG for 4 h at 4 °C followed by incubation overnight. The beads were washed three times with PBS and then eluted by boiling at 95 °C for 10 min. The immune complexes were separated by 10% sodium dodecyl sulfate-polyacrylamide gel electrophoresis (SDS-PAGE) and sent for MS analysis at Beijing Qinglian BioTech Co., Ltd. (Beijing, China).

### 2.4. Fusion Protein Expression and Purification

The pGEX-4T-1-KD plasmid was transformed into BL21 *Escherichia coli* competent cells. Cells were plated on Luria–Bertani (LB) agar with ampicillin (50 μg/mL) and incubated at 37 °C overnight. Single colonies were cultured in 5 mL LB broth with ampicillin, then diluted into 100 mL of fresh LB broth and grown to OD_600_ 0.6–0.8. Protein expression was induced by adding 1 mM isopropyl β-D-1-thiogalactopyranoside (IPTG; Catalog No. I8070, Solarbio, Beijing, China) followed by incubation at 28 °C for 12 h. Cells were harvested by centrifugation, washed three times with ice-cold PBS (pH 7.4), and resuspended in PBS (1 g wet weight/25 mL). Ultrasonication was performed on ice (40 min, 25% amplitude, 3 s on/5 s off) to lyse bacteria, and the lysate was centrifuged (12,000× *g*, 30 min, 4 °C) to isolate the soluble fraction. The supernatant was loaded onto a GST affinity column and eluted with 50 mM Tris-Cl (pH 8.0) and 10 mM reduced glutathione. Protein concentration was measured by the Bicinchoninic Acid (BCA) assay (Catalog No. KGP903, KeyGEN, Nanjing, China), followed by SDS-PAGE analysis and visualization with Coomassie Brilliant Blue staining (Catalog No. SW186, SEVEN Biotech, Beijing, China).

### 2.5. GST-Pulldown

Purified GST-ROCK1 kinase domain (GST-KD) fusion protein was mixed with lysates of MVC-infected WRD cells and incubated at 4 °C for 4 h under constant rotation. Subsequently, 100 μL of GST beads (Catalog No. DP201, TransGen) were added to the protein mixture and incubated overnight at 4 °C under constant rotation. After overnight binding, the mixture was placed on ice for 10 min, and the supernatant was removed. The beads were washed three times with ice-cold PBS to eliminate nonspecific interactions. Proteins bound to the beads were eluted by boiling in 5× loading buffer (Catalog No. CW 0027S, CoWin Biotech, Taizhou, China) at 95 °C for 10 min, followed by Western blot analysis to detect target proteins.

### 2.6. Cytotoxicity Assay

The RhoA inhibitor CCG-1423 (Catalog No. B4897, APExBIO, Houston, TX, USA) and the ROCK inhibitor Y27632 (Catalog No. A3008, APExBIO) were dissolved in dimethyl sulfoxide (DMSO; Catalog No. D8370, Solarbio) to prepare stock solutions at a concentration of 20 mM. The cell viability assay was performed using the CCK-8 assay kit (Catalog No. FC101, TransGen). WRD cells were seeded into 96-well plates at a density of 5 × 10^3^ cells per well. A series of working concentrations of CCG-1423 and Y27632 were prepared by diluting the stock solutions with the cell culture medium. Cells were treated with 100 μL of the medium containing different doses of the chemicals. The plates were then incubated in a 37 °C, 5% CO_2_ incubator for 24 h. After removal of the medium, 100 μL of freshly diluted CCK-8 solution was added, and the plates were returned to the 37 °C CO_2_ incubator for a 4 h reaction. The optical density at a wavelength of 450 nm (OD_450_) was measured using an enzyme-linked immunosorbent assay (ELISA) reader (Thermo Fisher Scientific). The formula for calculating cell viability is as follows: [(OD_sample_ − OD_blank_)/(OD_control_ − OD_blank_)] × 100, where OD_blank_ represents the optical density of the wells containing only the CCK-8 solution and the medium (without cells), and OD_control_ represents the optical density of the cell wells containing an equal amount of DMSO but no test chemicals.

### 2.7. siRNA Transfection

Three specific small interfering RNAs (siRNAs) targeting the ROCK1 gene were synthesized by Shanghai Genomics Technology Co., Ltd. (Shanghai, China), and their sequences are as follows: siROCK1, sense 5′-GAUACUAUAUGCAAAUGAAtt-3′, antisense 5′-UUCAUUUGCAUAUAGUAUCtt-3′. The negative control sequences of siRNAs (siNC) are as follows: the sense 5′-UUCUCCGAACGUGUCACGU-3′, antisense 5′-ACGUGACACGUUCGGAGAA-3′. WRD cells were seeded in 6-well plates at a density of 1 × 10^5^ cells per well and allowed to reach an appropriate confluence. Subsequently, the cells were transfected with ROCK1 siRNA at a final concentration of 100 nM. The siRNA was first mixed with 200 μL of Opti-MEM medium, followed by the addition of 10 μL of transfection reagent (Catalog No. FT201-01, TransGen). After thorough mixing, the solution was incubated at room temperature for 15 min to allow the formation of siRNA transfection reagent complexes. The entire mixture was then added to one well of the 6-well plate. The transfected cells were further cultured for an additional 24 h.

### 2.8. Plasmid Transfection

COS-1 cells were cultured in 100 mm dishes until they reached 80% confluence. The pCMV-HA-KD/-RBD/-PH-CRD, pCMV-HA-Occludin, or pXJ40-Flag-VP2 plasmid DNA was added to 500 μL of Opti-MEM medium (Catalog No. 31985070, Thermo Fisher Scientific) at a concentration of 10 μg per sample. Subsequently, 20 μL of transfection reagent was added to the plasmid containing Opti-MEM solution. The cells were incubated with the plasmid transfection reagent mixture and cultured for an additional 48 h to allow sufficient expression of the transfected genes.

### 2.9. Western Blot Analysis

Cells were washed three times with cold PBS and lysed on ice for 30 min using cell lysis buffer (Catalog No. KGP 701, KeyGEN) containing protease inhibitors. Bacterial lysates were subjected to ultrasonication, and the resulting supernatants were collected. Protein concentrations were quantified using the bicinchoninic acid (BCA) assay. Equal amounts of protein samples were separated by SDS-PAGE and subsequently transferred onto PVDF membranes (Catalog No. 0000229835, Millipore, Burlington, MA, USA). The membranes were blocked with 5% skim milk and incubated with primary antibodies at 4 °C overnight. The following primary antibodies were used: anti-ROCK1 (21850-1-AP; 1:10,000), anti-ZO2 (18900-1-AP; 1:1000), anti-TMED10 (67876-1-Ig; 1:10,000), anti-MLC2 (10906-1-AP; 1:2000), anti-Flag (66008-4-Ig; 1:10,000), anti-Occludin (66378-1-Ig; 1:10,000) and anti-β-actin (HRP-66009; 1:10,000), and all of them were purchased from Proteintech (Wuhan, China); anti-HA (AE105; 1:3000). The anti-GST (AE007; 1:3000) and anti-pMLC2 (AF6345; 1:2000) antibodies were obtained from ABclonal (Wuhan, China) and Affinity (Jiangsu, China), respectively. The anti-NS1 antibody (rabbit polyclonal, 18929-1; 1:1500) and anti-VP2 antibody (mouse monoclonal, 3C12-1; 1:1500) were developed in collaboration with Abmart (Shanghai, China). After washing, membranes were incubated with HRP-conjugated secondary antibodies at room temperature for 1 h (1:6000 dilution): Proteintech (goat anti-mouse/rabbit, RGAM001/SA00001-2) and Abbkine (mouse-anti-rabbit, A25022; goat-anti-mouse, A25012) (Wuhan, China). Immunoreactive bands were visualized using Enhanced ECL Prime (Catalog No. SW181-01, SEVEN Biotech) and chemiluminescent signals were captured using the Azure 300 Imaging System (San Jose, CA, USA). Relative protein expression levels were quantified using ImageJ software v1.46 (Bethesda, MD, USA).

### 2.10. Co-Immunoprecipitation (Co-IP) Assay

Co-transfected COS-1 cells or WRD cells infected with MVC were collected and solubilized in RRIPA buffer supplemented with protease inhibitors. Lysates were centrifuged at 12,000× *g* for 10 min at 4 °C, and supernatants were collected. As an input control, the supernatants were mixed with 5 × loading buffer and denatured at 95 °C for 10 min. For immunoprecipitation, lysates were incubated with 5 μg of anti-VP2, anti-ROCK1, anti-ZO2, anti-TMED10, anti-HA, anti-Flag, anti-GST, or anti-Occludin antibodies, or with species-matched IgG controls for 4 h at 4 °C. Protein A/G agarose beads (100 μL) (Catalog No. PR40025, Proteintech) were added and incubated overnight at 4 °C under rotation. Beads were washed five times with cold PBS, and bound proteins were eluted by boiling them in a 5× loading buffer for 10 min. Eluates were analyzed by SDS-PAGE and immunoblotting.

### 2.11. Immunofluorescence Assay (IFA) and Confocal Microscopy

The WRD cells infected with MVC or COS-1 cells transfected with pCMV-HA-KD and/or pXJ40-Flag-VP2 plasmids were seeded in confocal dishes (Catalog No. BDD011035, BIOFIL, Guangzhou, China), fixed with 4% paraformaldehyde in PBS at 4 °C for 15 min, permeabilized with 0.5% Triton X-100 for 20 min, and blocked with 5% normal goat serum (Catalog No. SL038, Solarbio) for 1 h at room temperature (RT). Primary antibodies (anti-ROCK1, anti-VP2, anti-HA, anti-Flag, and anti-Occludin; IFA: 1:100) were applied overnight at 4 °C. After PBS washes, cells were incubated with Alexa Fluor 488 (Catalog No. bs 0295G, Bioss, Beijing, China; IFA: 1:200) or Alexa Fluor 647 (Catalog No. bs-0296G, Bioss, IFA: 1:200) conjugated secondary antibodies for 1 h at RT. Nuclei were stained with 4′,6-diamidino-2-phenylindole (DAPI) (Catalog No. 62248, Thermo Fisher Scientific). Images were acquired using an LSM 800 confocal microscope (Carl Zeiss, Jena, Germany) and analyzed with ZEN software (v3.0, Carl Zeiss).

### 2.12. Quantity Real-Time PCR (qRT-PCR)

Total RNA from cells was isolated using the RNeasy Mini Kit (Catalog No. 12183020, Thermo Fisher Scientific). cDNA was synthesized from 1 μg of RNA using the PrimeScript RT reagent kit (Catalog No. RR047Q, Takara, Kyoto, Japan). qPCR reactions (20 μL total volume) contained 2 μL of cDNA, 10 μL of SGExcel FastSYBR Mix (Catalog No. 4444556, Thermo Fisher Scientific), and 0.4 μM of each primer (MVC VP2: F 5′-AGACGCTACTTCGCTACA-3′, R 5′-TACTGGACTGACATCATAACC-3′) [33]. Amplification was performed under the following conditions: 95 °C for 3 min, followed by 40 cycles of 95 °C for 3 s, 60 °C for 20 s, and 72 °C for 20 s. The calculation of viral genome copy numbers was performed as described earlier. Briefly, Hirt DNA was extracted, and viral supernatant was collected. The infectious clones of MVC (pI-MVC plasmid) [14] were quantified using a NanoDrop ND-P1000 spectrophotometer (Thermo Fisher Scientific, Waltham, MA, USA), and the copy number was calculated using the following formula: Copy number/mL = plasmid concentration (g/μL) × 6.022 × 10^23^/plasmid length (bp) × 660. A standard curve was constructed by performing a 10-fold serial dilution of pI-MVC plasmid. The copy number for each target was determined by comparing the Ct value of each sample with the standard curve. Non-template controls were included as negative controls for standard analysis. Data were collected and analyzed using Analytik Jena qTOWER^3^G (Jena, Germany).

### 2.13. Flow Cytometry for Membrane Permeability

Cells pretreated with inhibitors for 1 h were infected with MVC (MOI = 1) for 15 min, washed with PBS, and loaded with 2.5 μM Fluo-3 AM (Catalog No. S1056, Beyotime, Shanghai, China) in DMEM for 30 min at 37 °C, followed by 30 min at RT. After washing three times with PBS, the cells were analyzed by flow cytometry (Thermo Fisher Scientific). Fluorescence thresholds were set using unstained controls, with Fluo-3-positive cells defined as those exceeding the 95th percentile of the background signal. Debris and doublets were excluded by FSC/SSC gating and FSC-H vs. FSC-A gating, respectively. The data were analyzed using FlowJo v10.

### 2.14. Statistical Analysis

Statistical significance was assessed by one-way ANOVA followed by Tukey’s post hoc test (GraphPad Prism 5, San Diego, CA, USA). Significant differences were indicated as * *p* < 0.05, ** *p* < 0.01 and *** *p* < 0.001. Data represent the mean ± standard error of the mean (SEM) from at least three independent experiments. Figures were generated using Adobe Photoshop CC v2014 (San Jose, CA, USA).

## 3. Results

### 3.1. Identification of VP2-Interacting Host Proteins

The capsid proteins of non-enveloped viruses play a pivotal role as receptor-binding sites during host infection [34]. We hypothesize that host proteins interacting with VP2 may facilitate viral internalization during MVC infection. Here, we used IP coupled with MS to conduct a preliminary screening of host proteins interacting with VP2. Lysates from MVC-infected WRD cells were subjected to IP using anti-VP2 antibody, followed by SDS-PAGE separation and Coomassie Brilliant Blue staining. The gel was excised and processed for MS analysis, and datasets (Appendix A) were processed using the online Bioladder (https://www.bioladder.cn/web/#/pro/index, accessed on 1 December 2023). Triplicate experiments detected 218 candidate VP2-interacting proteins after subtracting IgG control signals (Figure 1A). The interacting proteins were categorized according to their cellular functional localization, distinct peptide count, coverage, and score (a composite metric derived from peptide-spectrum matching). Since our goal was to investigate how MVC infects WRD cells, ROCK1, ZO-2, and TMED10 emerge as prominent candidates based on their functional characteristics and comprehensive scores. Co-IP assays confirmed a specific interaction between VP2 and ROCK1 (Figure 1B,C), leading us to select ROCK1 for further investigation.

### 3.2. The Kinase Domain Mediates the Direct Interaction Between ROCK1 and VP2

As a key regulator of actin cytoskeletal dynamics, ROCK1 mediates cytoskeletal reorganization through the phosphorylation of actin-binding and associated structural proteins [35]. ROCK1, a member of the Ras superfamily, contains three conserved functional domains: the kinase domain (KD), the Rho-binding domain (RBD), and the pleckstrin homology-coiled-coil region (PH-CRD) (Figure 2A). To identify the specific ROCK1 domain that interacts with VP2, HA-tagged expression plasmids encoding individual domains (pCMV-HA-KD, pCMV-HA-RBD, and pCMV-HA-PH-CRD) were constructed. Each plasmid was co-transfected with pFlag-VP2 into COS-1 cells. Co-IP assays using specific antibodies revealed a direct interaction between the ROCK1-KD and VP2 (Figure 2B), whereas RBD and PH-CRD domains exhibited no binding capacity (Figure 2C,D).

Given the established role of the kinase domain in substrate recognition and phosphorylation [36,37], the interaction between the ROCK1-KD and VP2 was further validated through prokaryotic expression. Recombinant GST-KD fusion protein was successfully purified from *Escherichia coli* (Figure 3A). GST pulldown assays confirmed a direct physical interaction between the ROCK1-KD and VP2 (Figure 3B,C). Notably, antibody cross-reactivity between VP1 and VP2 was considered due to their shared sequence homology.

### 3.3. ROCK1 and VP2 Co-Localized in MVC-Infected WRD and Co-Transfected COS-1 Cells

To further observe and corroborate the associations of ROCK1 and VP2, we carried out immunofluorescence assay (IFA) in two experimental methods. (i) In WRD cells infected with MVC, endogenous co-localization of ROCK1 and VP2 was observed. In uninfected control cells, endogenous ROCK1 exhibited a diffuse cytoplasmic distribution at 6 h, 12 h, or 48 h post-infection (hpi) (Figure 4A, upper panel). Following MVC infection, ROCK1 co-localized with VP2 and translocated from the cytoplasm (12 hpi) to the nucleus (24 hpi) (Figure 4A). (ii) Further validation of co-localization was performed in COS-1 cells co-transfected with pCMV-HA-KD and pXJ40-Flag-VP2. As illustrated in Figure 4B, when pCMV-HA-KD was individually transfected, the exogenous ROCK1 kinase domain was predominantly localized in the cytoplasm. When pCMV-HA-KD was co-transfected with pXJ40-Flag-VP2, both proteins were diffusely distributed throughout the cell at 24 h post-transfection and migrated to the nucleus at 48 h post-transfection; during this period, the ROCK1 kinase domain co-localized with VP2. Taken together, these data strongly support that ROCK1 is associated with VP2. Although VP2 antibodies can recognize both VP1 and VP2 due to sequence homology, the transfection of pFlag-VP2 confirmed a specific interaction between ROCK1 and VP2.

### 3.4. MVC Activates pMLC via the RhoA/ROCK1 Signaling Pathway in WRD Cells

Several studies have reported that viral infections are closely associated with the activation of the RhoA/ROCK1 signaling pathway [23,28]. Given that the phosphorylation of myosin light chain 2 (pMLC2) is known to serve as a downstream target molecule of RhoA and ROCK1, we sought to determine whether MVC infection activates the RhoA/ROCK1/MLC2 signaling pathway in WRD cells. To verify this hypothesis, WRD cells were infected with MVC (MOI = 1) for different periods of time. Temporal activation patterns of proteins were analyzed by Western blot using specific antibodies targeting total RhoA, ROCK1, MLC2, and phosphorylated MLC2 (pMLC2). Our data demonstrated that in the early stage of MVC infection, MVC induced the activation of RhoA and ROCK1 as well as the early activation of pMLC2, and the peak activation was observed at 15 min post-infection, followed by a gradual decline (Figure 5).

To confirm that the RhoA/ROCK1/MLC2 signaling pathway is activated during the early stage of viral infection, WRD cells were pretreated for 1 h with inhibitors specific to RhoA (CCG-1423) or ROCK1 (Y27632), followed by incubation with MVC (MOI = 1) for 15 min. Western blot analysis was performed to assess the activation of ROCK1 and pMLC2. The concentration of each inhibitor was non-cytotoxic and did not affect WRD cell viability, as determined by the CCK-8 assay. The CCG-1423 at 4 μM and Y27632 at 20 μM were used as safe dosages in the subsequent experiments (Figure 6A,B). The expression levels of ROCK1 and the pMLC2/MLC2 ratio were decreased by 2-fold and 1.8-fold, respectively, in MVC-infected cells following pretreatment with the RhoA inhibitor CCG-1423. Similarly, ROCK1 expression and the pMLC2/MLC2 ratio were reduced by 1.8-fold and 3-fold, respectively, after treatment with Y27632 (Figure 6C,D). Next, we used siRNA targeting ROCK1 to further verify the regulatory effect of ROCK1 on MLC2. After being transfected with ROCK-siRNA, cells were infected with MVC for 15 min, and then protein expression was analyzed. As shown in Figure 6E, compared with MVC infection cells, ROCK1 and pMLC2/MLC2 ratio were decreased by 2.5-fold and 4-fold in pre-treated ROCK-siRNA cells, respectively. Taken together, these data suggest that pMLC2 activation in MVC-infected cells is dependent on the RhoA/ROCK1 signaling pathway.

### 3.5. VP2 Interacts with Occludin but Not with Claudin or ZO-1

Phosphorylation of MLC2 mediates actomyosin contraction, which is a well-established mechanism for TJ disassembly as documented in prior studies, and it plays a significant role in viral infections [38,39]. We wondered whether TJ proteins (Occludin, Claudin and ZO-1) play a role in viral infection and whether it interact with the VP2 protein. To this end, we used CO-IP to identify proteins that can interact with VP2 in WRD cells infected with MVC. As shown in Figure 7A,B, VP2 interacted only with Occludin, but not with Claudin or ZO-1 proteins. To further validate this interaction, we conducted mutual Co-IP assays in COS-1 cells co-transfected with pCMV-HA-Occludin and pXJ40-Flag-VP2. The results further confirmed that Occludin interacts with VP2 (Figure 7C,D).

To understand the correlation between the RhoA/ROCK1/MLC2/Occludin pathway and MVC infection, we analyzed the protein expression levels of Occludin in WRD cells infected with MVC and evaluated the effects of RhoA and ROCK1 inhibitors on its expression. As expected, the upregulation of Occludin expression correlated with the early activation of pMLC2, which was induced by MVC and peaked at 15 min post-infection (Figure 8A). However, Occludin expression levels decreased by 1.7-fold and 3.4-fold following treatment with CCG-1423 and Y27632, respectively (Figure 8B,C). Consistently, ROCK1 knockdown via siRNA further reduced Occludin levels by 1.4-fold compared to those in the siNC group (Figure 8D). According to the results shown above, the RhoA/ROCK1 signaling pathway plays a significant role in regulating the expression of Occludin during the early stage of MVC infection.

### 3.6. Inhibition of the RhoA/ROCK1/MLC2 Signaling Pathway Reduces MVC Viral Genome Copies and Viral Protein Expression

The RhoA/ROCK/MLC signaling pathway was activated during the early stage of MVC infection, therefore, we investigated whether RhoA/ROCK1 signaling pathway is involved in MVC infection and entry. (i) The WRD cells were pretreated for 1 h with specific inhibitors targeting RhoA (CCG-1423) or ROCK1 (Y27632), followed by MVC infection for 6 h, 12 h, and 18 h. The results showed that, compared with the cells infected with MVC alone, CCG-1423 and Y27632 reduced the viral genome copy numbers in the culture supernatants of MVC-infected cells by 28%, 27%, and 20%, respectively (Figure 9A), or by 16%, 17%, and 34% (Figure 9C). Correspondingly, intracellular Hirt DNA levels declined by 12%, 56%, and 22% (Figure 9B) or by 20%, 81%, and 24% (Figure 9D) in CCG-1423 or Y27632 pre-treated cells. Furthermore, VP2 and NS1 proteins decreased by 1.8-fold and 1.6-fold (Figure 9E) or by 1.6-fold and 1.1-fold at 18 h post-infection (Figure 9F). (ii) To further examine the role of ROCK1 in the early stage of MVC infection, ROCK1 knockdown using siRNA was performed in WRD cells. As shown in Figure 9G, MVC NS1 levels decreased by 0.6-fold as early as 8 h post-infection. These results suggested that early activation of the RhoA/ROCK1/MLC2 signaling pathway by MVC significantly influenced the production of MVC progeny virions.

### 3.7. MVC-Induced Early Activation of the RhoA/ROCK1/MLC2 Pathway Modifies Occludin Protein Distribution and Cell Membrane Permeability

During basolateral viral invasion, TJ proteins are relocated from the cell surface to the cytoplasm, a process that has been reported in several studies and is associated with MLC2 phosphorylation in epithelial cells [23,29]. Thus, we questioned whether the internalization of Occludin in MVC-infected cells could be inhibited by blocking the RhoA/ROCK1/MLC2 pathway. To verify the hypothesis, WRD cells were pretreated with RhoA inhibitor (CCG-1423) or ROCK inhibitor (Y27632) and then infected with MVC. In comparison to the untreated control group, MVC-infected WRD cell monolayers showed a significant decrease in the cytoplasmic rearrangement of Occludin. These observations demonstrated that inhibition of the RhoA/ROCK1/MLC2 signaling pathway partially reduced the translocation of Occludin from the junctional area to the cytoplasm (Figure 10A).

To further evaluate the effects of MVC-induced RhoA/ROCK1 activation on TJ integrity, we measured cytoplasmic calcium levels using Fluo-3 fluorescence. The intracellular calcium level is an indicator of cell membrane permeability [40,41]. MVC infection resulted in a 3-fold increase in intracellular calcium ion levels compared to uninfected cells. However, pretreatment with ROCK1 or RhoA inhibitors effectively attenuated this effect, resulting in significantly lower intracellular calcium levels (Figure 10B). The increase in Fluo-3 detected by flow cytometry was consistent with the IFA results (Figure 10C). Collectively, our findings suggested that the RhoA/ROCK1/MLC2 signaling pathway plays a crucial role in regulating actomyosin contraction and maintaining TJ integrity, with its activation leading to TJ disruption, Occludin exposure, and facilitated viral entry, as illustrated in Figure 10D.

## 4. Discussion

Viral attachment to host cells represents a critical step in the infection process, with host range selection and infection efficiency primarily determined by structural protein-mediated receptor recognition and signaling regulatory networks [42]. As a core component of the viral capsid, the structural protein of virus (VP) not only mediates viral invasion through interactions with classical receptors, such as integrins and sialic acid-binding proteins [43,44,45], but also exhibits functional diversity by regulating host cell signaling pathways and antiviral responses. However, the specific signaling pathways through which MVC capsid proteins mediate viral entry remain unclear. In this study, we revealed for the first time that the kinase domain of the host protein ROCK1 directly interacts with the MVC VP2 protein, playing a key role in optimizing MVC infection. During the early stage of viral infection, MVC activates the RhoA/ROCK1/MLC2 signaling pathway. Subsequently, the phosphorylation of MLC2 leads to the contraction of TJ ring, enabling the VP2 protein to bind to the exposed tight junction protein Occludin, thus promoting viral invasion. Moreover, we further identified that RhoA/ROCK1 inhibitors effectively block MVC infection.

ROCK1 is a RhoA-GTP-activated serine/threonine kinase that primarily regulates cell contraction, vesicle transport, and cell barrier integrity [24,46]. It consists of three functional domains: an N-terminal KD that phosphorylates substrates, a central RBD mediating RhoA-GTP interaction to control activation, and a C-terminal region containing PH-CRD directing subcellular localization [47]. The N-terminal kinase domain of ROCK1 plays a biological function, phosphorylating key substrates such as MLC, LIM kinase (LIMK), and cofilin, thereby regulating cytoskeletal dynamics, cell contractility, and membrane tension. In this study, by using the method of IP coupled with MS, we first screened out ROCK1 from 218 candidate proteins that interact with MVC VP2 protein and further identified that the kinase domain of ROCK1 is involved in the interaction (Figure 3B). This interaction suggests that the virus may bind to the ROCK1 kinase domain, thereby enhancing ROCK1 activation and promoting viral entry. Additionally, ROCK1 kinase activity can be regulated through autophosphorylation, and its aberrant activation is closely linked to cell barrier disruption, increased contractility, and abnormal vesicle transport, all of which may facilitate viral infection. Notably, ROCK1’s role in viral infection likely operates through two distinct mechanisms. Firstly, ROCK1-mediated phosphorylation of MLC2 induces actomyosin contraction, facilitating viral membrane fusion and disrupting epithelial barriers. Muhammad Sharif et al. reported that porcine sapovirus (PSaV) compromises tight junction integrity in polarized LLC-PK cells by triggering the RhoA/ROCK/MLC signaling pathway, promoting viral invasion [23]. Similarly, Mahmoud Soliman et al. found that species A rotavirus (RVA) perturbs tight junction stability in polarized MDCK cells through the same signaling cascade, causing tight junction protein redistribution and actomyosin ring contraction, thereby enhancing viral penetration [28]. Second, ROCK1 regulates actin cytoskeleton remodeling by activating the LIMK/cofilin pathway, promoting cell membrane invagination to form clathrin-mediated endocytic structures and accelerating viral internalization. This mechanism is similar to how prion-like protein aggregates, including mutant SOD1, tau, and α-synuclein, exploit the RhoA to cofilin signaling pathway to remodel actin and facilitate cellular entry [48]. Moreover, HIV activates Rho family GTPases (including RhoA, Cdc42, and Rac1) through CD4 and CCR5 receptor signaling, with RhoA/ROCK promoting HIV infection of CD4+ T cells by regulating cytoskeletal dynamics [49]. In our study, we demonstrated that the RhoA/ROCK1/MLC2 signaling pathway is activated during the early stages of MVC infection, particularly the phosphorylation of MLC2. Next, we employed two chemical inhibitors, CCG-1423 and Y27632, to further investigate the impact of the RhoA/ROCK1/MLC2 signaling pathway on viral infection. CCG-1423 is a small-molecule inhibitor that significantly reduces stress fiber formation and focal adhesion stability by inhibiting RhoA signaling, thereby preventing ROCK activation and subsequent phosphorylation of MLC2 [50]. Y27632, a competitive inhibitor, blocks ROCK’s kinase activity by binding to its ATP-binding site, thereby inhibiting downstream effects such as cytoskeletal remodeling and TJ regulation [51]. These inhibitors have been extensively applied in the studies of viral infections. Several reports have shown that Y27632 reduces the ability of HIV, influenza virus, and coronaviruses to mediate viral entry by modulating the cytoskeleton [52,53,54]. Here, we showed that inhibitors of RhoA or ROCK, as well as siRNA targeting ROCK1 significantly decrease MVC genome copy numbers and viral protein expression in WRD cells infected with MVC (Figure 9).

Given that the activation of the RhoA/ROCK1/MLC2 signaling pathway mediates TJ disassembly, we observed that this process occurs during the early stage of MVC infection. Specifically, RhoA-GTP activation and its downstream effector ROCK1 promote MLC2 phosphorylation, which triggers actin-myosin ring contraction and leads to the TJ disassembly. The TJ disruption weakens the cellular barrier, exposing proteins such as Occludin, Claudin, and ZO-1, which facilitate viral entry by enabling interaction with target molecules [55,56,57]. Occludin acts as a downstream target of the RhoA/ROCK/MLC signaling pathway, and its function and stability are regulated by this pathway, playing a crucial role in TJ remodeling and barrier permeability modulation. As a transmembrane protein, Occludin is as an important component of TJs in epithelial and endothelial cells, maintaining cellular barrier integrity through its structure, which comprises four transmembrane helices (TMDs), two extracellular loops (EL1 and EL2), an N-terminal, and a C-terminal cytoplasmic domain [58]. Beyond its structural role, Occludin acts as a dual-function regulator during viral infections: it serves as both a molecular gateway for viral entry and a mediator of barrier disruption, thereby amplifying viral spread. Lavie et al. reported that HCV recruits Occludin away from tight junctions, slowing viral particle motility and promoting internalization [59]. Similarly, Liu et al. demonstrated that the second extracellular loop (EL2) of Occludin is critical for HCV entry by mediating its interaction with Dynamin II, which facilitates virus internalization via a Dynamin-dependent endocytic pathway [60,61]. Occludin also maintains hepatocyte polarity and enhances infection efficiency by promoting the co-localization of the CD81/Occludin receptor complex. Luo et al. showed that, during PEDV infection, Occludin undergoes redistribution in porcine intestinal epithelial cells (IPEC-J2), facilitating viral entry through macropinocytosis [29]. Likewise, rotavirus infection disrupts Occludin-mediated TJ integrity in intestinal epithelial cells, increasing permeability and viral dissemination [28]. Furthermore, Occludin directly interacts with the SARS-CoV-2 spike protein, facilitating viral entry via the endosomal pathway and promoting cell-to-cell fusion [62]. In our study, Co-IP assays confirmed that MVC VP2 interacts specifically with Occludin but not with claudin or ZO-1. This interaction suggests that MVC may exploit Occludin-mediated mechanisms for viral entry. Meanwhile, during the early stage of MVC infection, Occludin expression was upregulated. These observations suggest that Occludin may not only function as a receptor but also participate in the regulation of barrier properties during the viral invasion process. Additionally, RhoA and ROCK inhibitors significantly reduced pMLC2 activation during early MVC infection, preventing TJ dissociation and partially restoring Occludin localization (Figure 10A,B). This restoration correlated with reduced membrane permeability, indicating partial recovery of TJ barrier function.

Although MVC infection activates the RhoA/ROCK1/MLC2 signaling pathway and significant effects on viral infection, our study still has some limitations. In WRD cells, as MVC infection progressed from 6 h to 24 h, ROCK1 and VP2 gradually relocated from the cytoplasm to the nucleus. These findings might imply that ROCK1 regulate viral replication or assembly in the nucleus through its interaction with VP2, further studies are needed to confirm this hypothesis. In addition, the Occludin was transiently upregulated in the early stage of MVC infection, and whether it acts as an assisting receptor for MVC entry into cells still needs to be further investigated.

In conclusion, our study demonstrated that the host protein ROCK1 acts as a binding partner for MVC VP2. During the early stage of MVC infection, the RhoA/ROCK1/MLC2 signaling pathway is activated, resulting in TJ dissociation, Occludin exposure, and an increase in the permeability of the cell membrane, all of which facilitated MVC entry into host cells. These findings help to elucidate a preliminary but significant mechanism linking the RhoA/ROCK1 signaling pathway to MVC infection. It can contribute to advancing our understanding of the mechanisms and pathogenicity of MVC infection, as well as inform the future development of novel antiviral therapies.

## Figures and Tables

**Figure 1 microorganisms-13-00695-f001:**
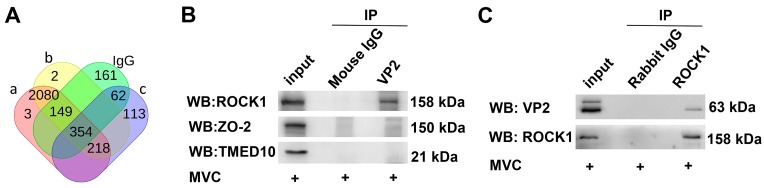
Screening and identification of VP2 and host cell proteins. (**A**) Peptide analysis of MS in MVC-infected cells. WRD cells infected with MVC (MOI = 1) for 48 h were collected. Lysates were subjected to IP using an anti-VP2 antibody, followed by SDS-PAGE separation and Coomassie Brilliant Blue staining. The gel was analyzed by MS. After subtraction of IgG control datasets from triplicate experimental replicates, 218 candidate VP2-interacting host proteins were screened. a, b, and c represent independent MS datasets. (**B**) Co-IP of VP2-interacting proteins. WRD cells infected with MVC (MOI = 1) were lysed, and Co-IP was performed using a mouse anti-VP2 antibody, detected by Western blot with rabbit anti-ROCK1, rabbit anti-ZO-2, and mouse anti-TMED10 antibodies. Mouse IgG served as a negative control. (**C**) Similar to (**B**), anti-ROCK1 antibody was used instead of the anti-VP2 antibody. Rabbit IgG served as a negative control.

**Figure 2 microorganisms-13-00695-f002:**
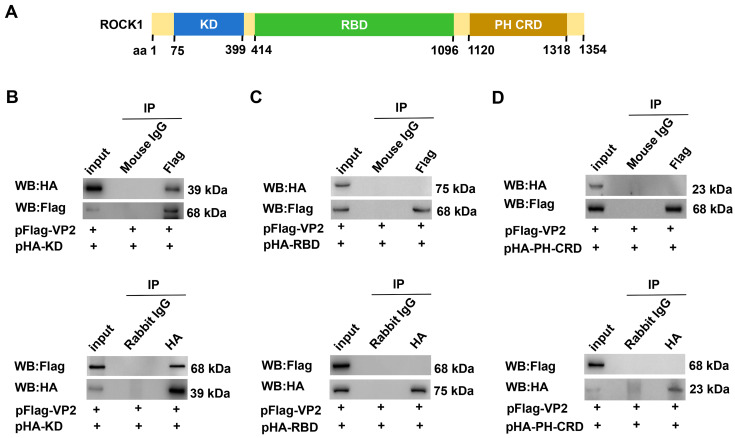
The interaction between ROCK1 and VP2 is mediated through the kinase domain. (**A**) A schematic representation of the ROCK1 domains used in the Co-IP analysis, with amino acid positions annotated. (**B**) Interaction analysis between the ROCK1 kinase domain and VP2 protein. Plasmids pCMV-HA-KD (pHA-KD) and pFlag-VP2 were co-transfected into COS-1 cells as indicated. Co-IP assays were performed using a mouse anti-Flag or rabbit anti-HA antibodies to detect protein interactions. Mouse or rabbit IgG served as a negative control. (**C**,**D**) Similar to (**B**), COS-1 cells were co-transfected with either pCMV-HA-RBD (pHA-RBD) or pCMV-HA-PH-CRD (pHA-PH-CRD) plasmids, along with pXJ40-Flag-VP2 (pFlag-VP2), as indicated. Co-IP assays were performed under the same conditions as in panel (**B**).

**Figure 3 microorganisms-13-00695-f003:**
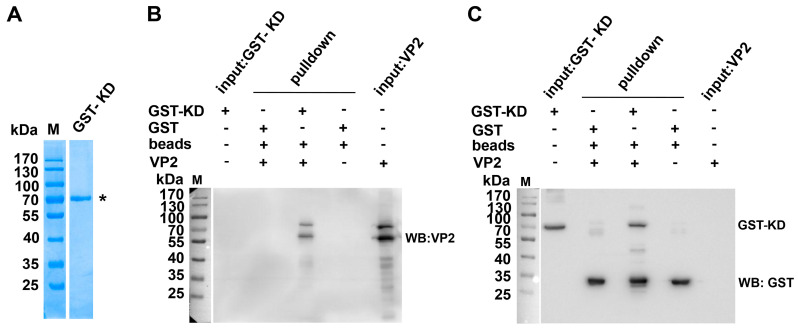
Verification of the interaction between the kinase domain of ROCK1 and VP2 in vitro. (**A**) Purification of GST-KD fusion protein. The GST-KD fusion protein was eluted from an affinity chromatography column using reduced glutathione as the elution buffer. The eluted protein was then analyzed by SDS-PAGE (10%) and visualized using Coomassie Brilliant Blue staining. The asterisk (*) denotes GST-KD. (**B**,**C**) GST pulldown assay. Purified GST-KD fusion protein was incubated with lysates from MVC-infected WRD cells, followed by GST pulldown assays to assess protein interactions.

**Figure 4 microorganisms-13-00695-f004:**
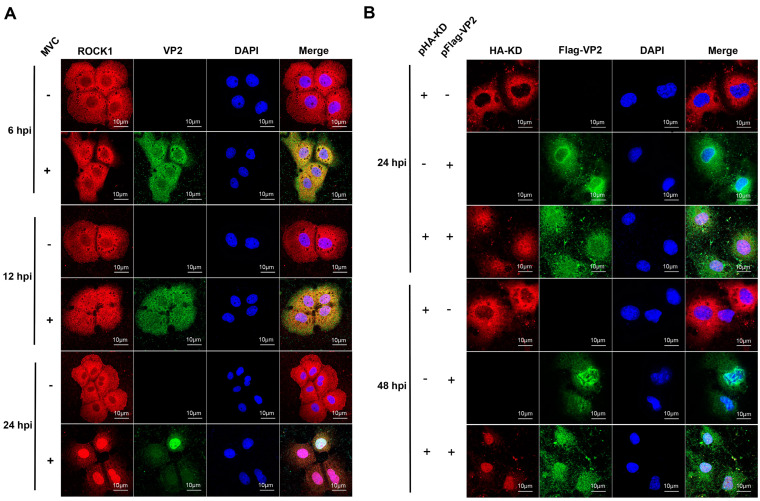
Co-localization of ROCK1 and VP2 in WRD and COS-1 cells. (**A**) Subcellular localization of ROCK1 (red) and MVC VP2 (green) in WRD cells infected with MVC (MOI = 1) was examined using confocal microscopy at 6, 12, and 24 h post-infection. MVC-infected and uninfected WRD cells were fixed and subjected to IFA using rabbit anti-ROCK1 (1:100) and mouse anti-VP2 (1:100) antibodies, followed by incubation with Alexa Fluor 488 and/or Alexa Fluor 647-conjugated secondary antibodies (1:200). DAPI (blue) was used for nuclear staining. (**B**) In COS-1 cells, pCMV-HA-KD (red) and pXJ40-Flag-VP2 (green) were transfected either individually or in combination. The subcellular localization of the expressed proteins was determined at 24 and 48 h post-transfection. Transfected COS-1 cells were fixed and incubated with rabbit anti-HA (1:100) and mouse anti-Flag (1:100) antibodies, followed by the same secondary antibodies as in (**A**). Scale bars: 10 μm.

**Figure 5 microorganisms-13-00695-f005:**
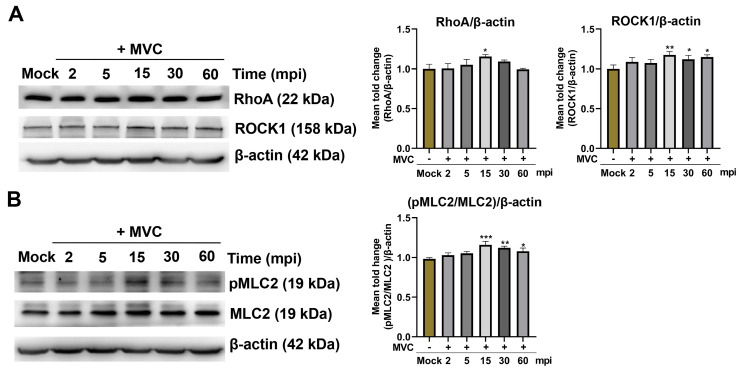
Early MVC infection activates the RhoA/ROCK1 signaling pathway and induces MLC2 phosphorylation in WRD cells. MVC-infected WRD cells (MOI = 1) were harvested at the indicated time points. (**A**) Western blot analysis of RhoA and ROCK1 expression levels. Cell lysates were analyzed by Western blot using rabbit anti-RhoA and rabbit anti-ROCK1 antibodies to detect the expression levels of RhoA and ROCK1. The term ‘mpi’ (minutes post-infection) denotes the time points post-infection. (**B**) Western blot analysis of MLC2 phosphorylation. The pMLC2 and total MLC2 were detected by Western blot using rabbit anti-MLC2 and rabbit anti-pMLC2 antibodies. β-actin was used as a loading control. The data were normalized to β-actin and are presented as the mean ± SEM from three independent experiments. Statistical analysis was performed using one-way ANOVA compared to the Mock group. The statistical analysis results are shown in the graphs on the right-hand side of each Western blot. Significance levels are denoted as * *p* < 0.05; ** *p* < 0.01; *** *p* < 0.001.

**Figure 6 microorganisms-13-00695-f006:**
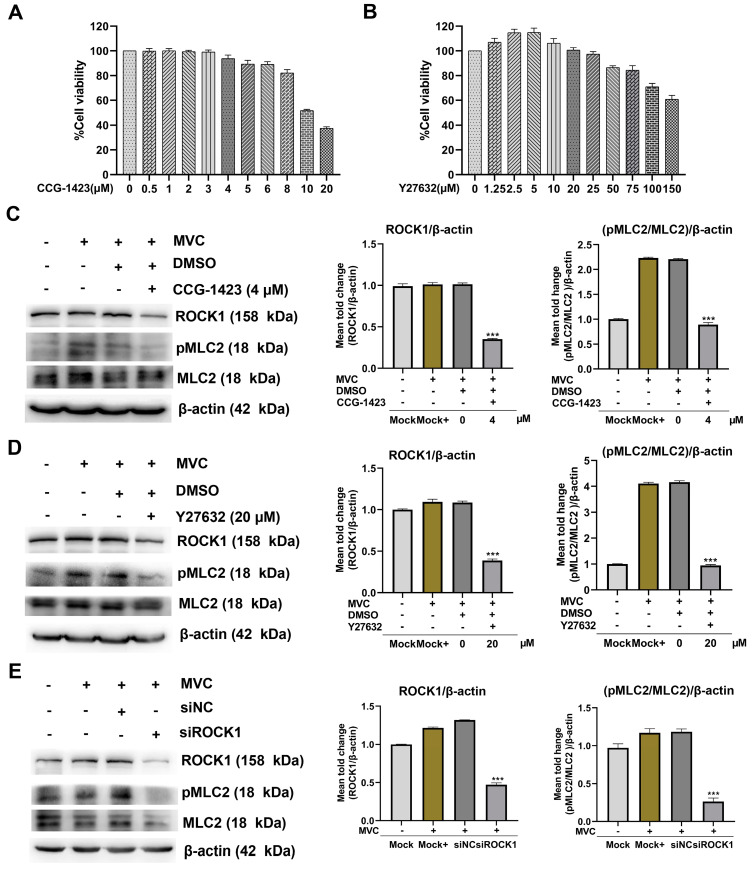
Inhibition of the RhoA/ROCK1 signaling pathway reduces pMLC2 expression during the early stage of MVC infection in WRD cells. (**A**,**B**) Assessment of cytotoxic effects of inhibitors using the CCK-8 assay. WRD cells seeded in 96-well plates were incubated with different concentrations of CCG-1423 or Y27632 at 37 °C for 24 h, followed by incubation with CCK-8 reagent at 37 °C for 4 h. Cell viability was evaluated by measuring OD450 using an ELISA reader. (**C**,**D**) Western blot analysis of ROCK1 and pMLC2/MLC2 expression following inhibitor treatment. WRD cells were pretreated with 4 μM of the RhoA inhibitor CCG-1423 (**C**) or 20 μM of the ROCK1 inhibitor Y27632 (**D**) at 37 °C for 1 h, followed by replacement with fresh culture medium. The cells were then infected with MVC (MOI = 1), and cell lysates were collected at 15 min post-infection (mpi). The expression levels of ROCK1 and pMLC2/MLC2 were analyzed by Western blot using rabbit anti-ROCK1, rabbit anti-MLC2, and rabbit anti-pMLC2 antibodies. (**E**) siRNA-mediated knockdown of ROCK1. ROCK1 expression was knocked down in WRD cells using siRNA, while a scrambled siRNA (siNC) was used as a negative control. Cell lysates were analyzed by Western blot to assess the expression levels of ROCK1 and pMLC2/MLC2. Data from the siROCK1 group were compared with those of the siNC group. The data were normalized to β-actin and are presented as the mean ± SEM from three independent experiments. Statistical significance was determined using one-way ANOVA followed by Tukey’s post hoc test, compared to the Mock+ group. The statistical analysis results are shown in the graphs on the right-hand side of each Western blot. Significance levels are denoted as *** *p* < 0.001.

**Figure 7 microorganisms-13-00695-f007:**
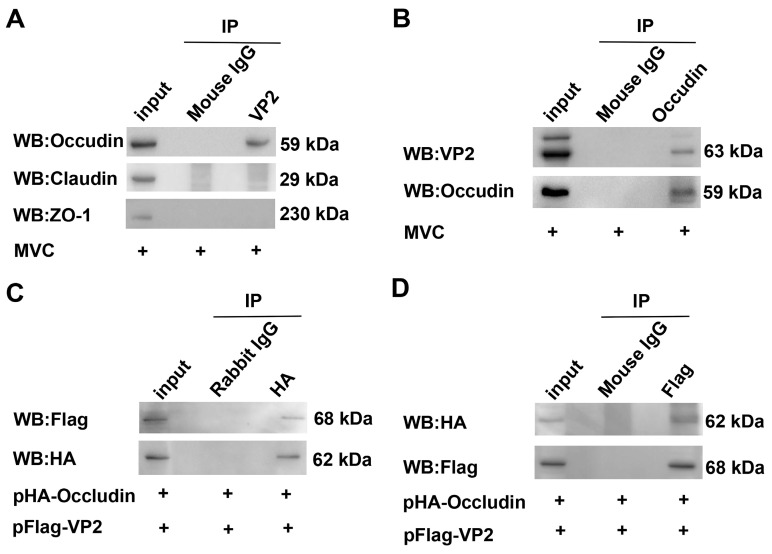
Identification of the interaction between Occludin and VP2. (**A**) WRD cells infected with MVC (MOI = 1) were lysed and subjected to co-immunoprecipitation (Co-IP) using an anti-VP2 antibody. The Co-IP eluates were analyzed by Western blot using mouse anti-Occludin, rabbit anti-Claudin, and mouse anti-ZO-1 antibodies. Mouse IgG served as a negative control. (**B**) Similar to (**A**), but an anti-Occludin antibody was used for Co-IP instead of the anti-VP2 antibody. The Western blot analysis was then performed using anti-VP2 and anti-Occludin antibodies. (**C**,**D**) Plasmids pCMV-HA-Occludin and pXJ40-Flag-VP2 were co-transfected into COS-1 cells. Co-IP assays were performed using either a mouse anti-Flag or rabbit anti-HA antibody to detect potential protein–protein interactions. Rabbit or mouse IgG was included as a negative control.

**Figure 8 microorganisms-13-00695-f008:**
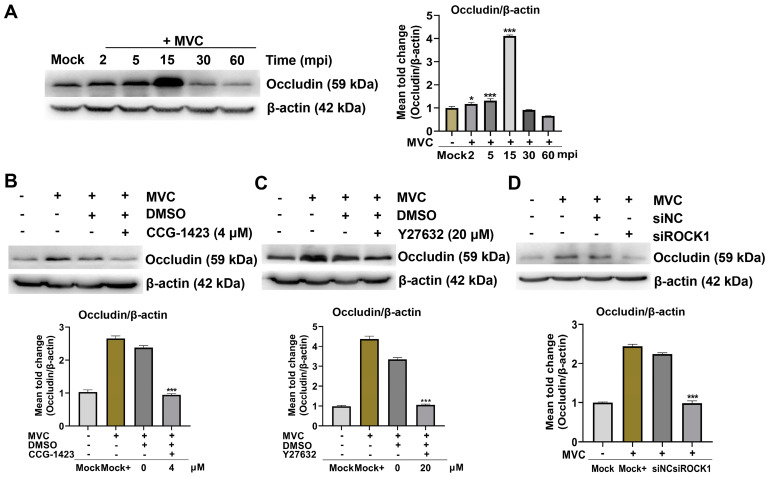
Occludin expression is upregulated in WRD cells during early MVC infection. (**A**) Western blot analysis of Occludin expression in MVC-infected WRD cells. The WRD cells were infected with MVC (MOI = 1) and harvested at the designated time points. (**B**,**C**) Effect of RhoA/ROCK1 inhibition on Occludin expression. Pretreated WRD cells with either 4 μM CCG-1423 or 20 μM ROCK1 inhibitor Y27632 at 37 °C for 1 h, followed by replacement with fresh culture medium. The cells were then infected with MVC (MOI = 1), and cell lysates were collected at 15 min post-infection (mpi). (**D**) Effect of ROCK1 knockdown on Occludin expression. Transfected WRD cells with 100 nM siRNA targeting ROCK1 for 24 h, followed by infection with MVC (MOI = 1) for 15 min. siNC was used as a negative control. The expression levels of Occludin were evaluated by Western blot analysis using mouse anti-Occludin antibodies. The data were normalized to β-actin and are presented as the mean ± SEM from three independent experiments. Statistical significance was determined using one-way ANOVA followed by Tukey’s post hoc test, compared to the Mock or Mock+ group. * *p* < 0.05; *** *p* < 0.001.

**Figure 9 microorganisms-13-00695-f009:**
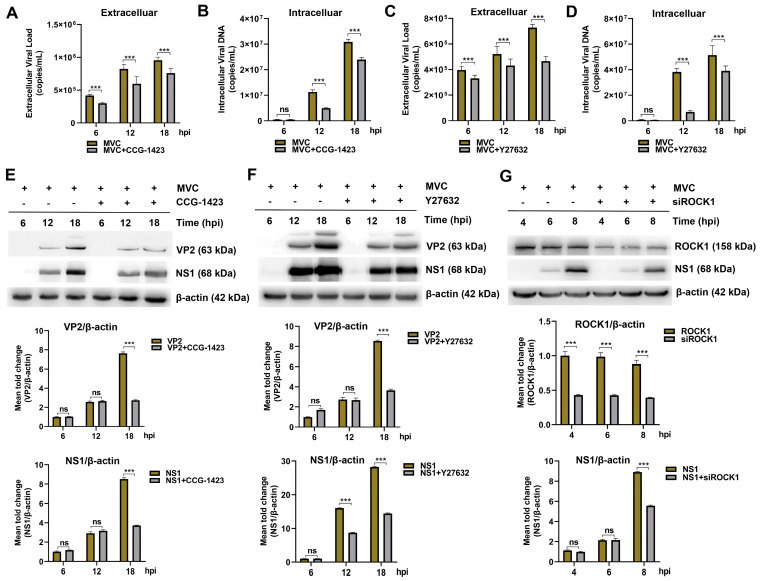
Inhibition of the RhoA/ROCK1/MLC2 signaling pathway affects MVC progeny viral genome copies and protein expression. WRD cells were pretreated with CCG-1423 or Y27632 at 37 °C for 1 h, followed by infection with MVC (MOI = 1). (**A**–**D**) qRT-PCR analysis of viral genome copies. Extracellular viral particles (from infected supernatant) and intracellular viral DNA (from Hirt DNA) were harvested at 6 h, 12 h, and 18 h post-infection (hpi). The extracellular and intracellular viral genome copy numbers were measured by qRT-PCR. (**E**–**G**) Western blot analysis of viral protein expression. WRD cells were pretreated with CCG-1423 (**E**) or Y27632 (**F**) or siRNA targeting ROCK1 (**G**) at the indicated concentrations for 1 h, followed by infection with MVC (MOI = 1). Western blot analysis was performed at 6 h, 12 h, and 18 h post-infection (hpi) to assess the expression levels of viral proteins (VP2 and NS1), and at 4, 6, and 8 hpi to evaluate the expression of ROCK1 and NS1, using rabbit anti-NS1, mouse anti-VP2, and rabbit anti-ROCK1 antibodies. β-actin was used as a loading control. The data were normalized to β-actin and are presented as the mean ± SEM from three independent experiments. Statistical significance was determined using two-way ANOVA, compared to the infection control group. The statistical analysis results are shown as graphs below each Western blot. *** *p* < 0.001; ns, non significant.

**Figure 10 microorganisms-13-00695-f010:**
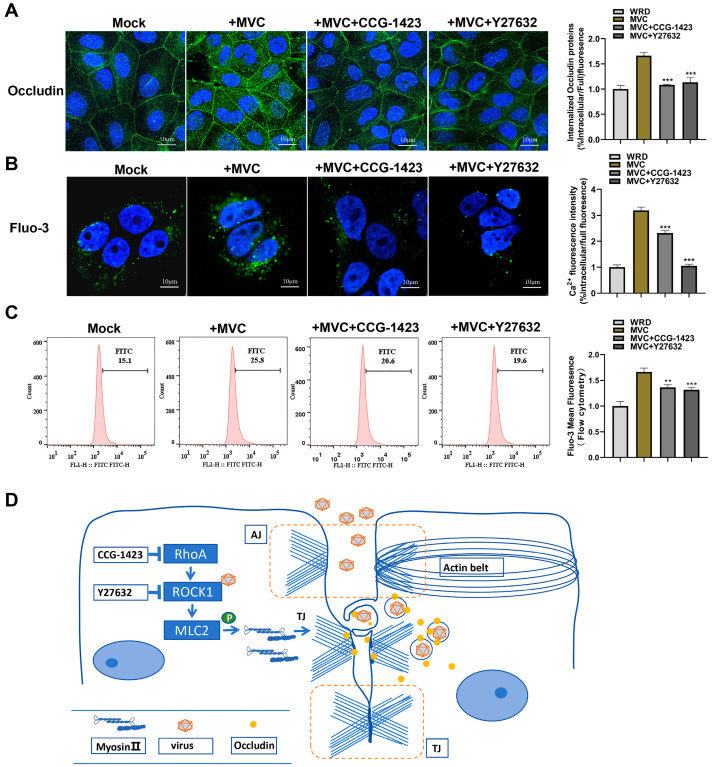
Activation of the RhoA/ROCK1/MLC2 signaling pathway alters Occludin distribution and increases cell membrane permeability during early MVC infection. WRD cells were pretreated with CCG-1423 or Y27632 at 37 °C for 1 h and then infected with MVC (MOI = 1) for 15 min. (**A**) IFA of Occludin internalization. Cells were fixed, permeabilized, and stained using mouse anti-Occludin antibody and an AlexaFluor 488-conjugated secondary antibody, then observed under a confocal microscope. Internalized Occludin was quantified as the percentage of intracellular fluorescence/total fluorescence. Scale bars: 10 μm. (**B**,**C**) Measurement of cytoplasmic calcium levels. The fluorescent calcium indicator Fluo-3 (2.5 μM) was used to observe cytoplasmic calcium ion concentration changes through laser confocal microscopy and flow cytometry. Statistical significance was determined using one-way ANOVA followed by Tukey’s post hoc test. ** *p* < 0.01; *** *p* < 0.001. (**D**) Schematic representation of MVC entry regulation via the RhoA–Occludin pathway. AJ: Adherens Junctions; TJ: Tight Junctions.

**Table 1 microorganisms-13-00695-t001:** Primer sequences are used to construct plasmids.

Primers ^a^	(bp)	Cloning Sites	Sequences (5′-3′) ^b^
pCMV-HA-KD F	972	*Sal*I	CGCGTCGACGATTATGAAGTAGTGAAGGTG
pCMV-HA-KD R		*Xho*I	TCCCTCGAGTGTAATTACTGTTCACCCTAC
pCMV-HA-RBD F	2046	*Sal*I	CGCGTCGACCGAAGTAGCCCCAATGTGGATA
pCMV-HA-RBD R		*Xho*I	TCCCTCGAGGTCCAAAAGTTTGGCACGCAGT
pCMV-HA-PH-CRD F	594	*Sal*I	CGCGTCGACAGAATTGAAGGTTGGCTTTCAA
pCMV-HA-PH-CRD R		*Xho*I	TCCCTCGAGTCTTCTTTACCAAAATGTGTTA
pCMV-HA-Occludin F	1566	*Sal*I	CGCGTCGACCAGGTTGGCTTATTTTGGGGA
pCMV-HA-Occludin R		*Xho*I	TCCCTCGAGTGCAAAGTTCACCGTGGGACC
pGEX-4T-1-KD F	972	*Eco*RI	GCGGAATTCGATTATGAAGTAGTGAAGGTG
pGEX-4T-1-KD R		*Xho*I	GCGCTCGAGTGTAATTACTGTTCACCCTAC

^a^ F: forward primer, R: reverse primer. KD: kinase domain; RBD: Rho-binding domain; PH CRD: carboxyl-terminal region contains a pleckstrin homology domain. ^b^ The underlined letters indicate restriction enzyme cleavage sites for cloning.

## Data Availability

The original contributions presented in this study are included in the article/Appendix A. Further inquiries can be directed to the corresponding author.

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
