# Peer review of "The Minute Virus of Canines (MVC) Activates the RhoA/ROCK1/MLC2 Signal Transduction Pathway Resulting in the Dissociation of Tight Junctions and Facilitating Occludin-Mediated Viral Infection"

_microorganisms, 2025, doi:10.3390/microorganisms13030695_

Round 1

Reviewer 1 Report

Comments and Suggestions for Authors

The introduction is well written, appears well structured and is considered to be written in a scientifically rigorous manner, but it is believed that there are some sections that could be improved. From lines 38 to 45, it is described how the pathogen can affect puppies and what diseases it can cause, affecting, as the introduction shows, different organs. It would be interesting to learn more about how and why understanding this virus is so crucial and what impact it can have on public health and veterinary practice. Within the introduction there are a number of different sections, including one describing the MVC virus, another addressing the pathological effects of the virus on dogs and another introducing the WRD cell system and ROCK1 proteins, in which regard it would be necessary to merge the sections with more sentences between each section. The materials and methods section is well detailed, there are some parts in the different sections that could be improved. It is requested that the authors be more precise with respect to the experimental conditions and methods used e.g. with respect to transfection steps and cell cultures. What is meant by standard conditions. The section on transfection is clear and correctly written, it would be necessary to write exactly how the treatments are chosen and how this may affect the experiment.

The discussion describes all the different aspects that have been studied. More details should be included with respect to the molecular mechanism of TJs disruption. Furthermore, some studies should be added with respect to the modulation of the RhoA/ROCK pathway. A better definition of the future goal is requested.

Author Response

Dear Reviewer,

Thank you for the editor and reviewers’ valuable comments on our manuscript entitled “RhoA/ROCK1/MLC2 Induced Positively by MVC Modulates Tight Junction Dissociation and Facilitates Occludin-Mediated Viral Infection” (Manuscript ID: microorganisms-3468333). We thoroughly revised the manuscript based on the comments.

Reviewer comments:

Comment 1: From lines 38 to 45, the pathogen is described as affecting puppies and causing diseases, affecting, as the introduction shows, different organs. It would be interesting to learn more about why understanding this virus is crucial and what impact it can have on public health and veterinary practice.

Response 1: Thank you for your insightful suggestion. We have added the following content to the manuscript, from line 55 to line 66. “ For host cell recognition, the capsid attaches to specific receptor(s) on the cell surface. Although Blackburn et al.. reported that BPV specifically binds to O-linked α2,3-linked sialic acids on GPA [1], the cell entry mechanisms mediated by the capsid proteins of other bocavirus species remain poorly characterized. The full length of the MVC genome exhibits 43% and 52.6% identity with the genomes of BPV and HBoV, respectively, sharing similarities in genomic organization and pathogenic mechanisms [2]. Based on the fact that the MVC infection of Walter Reed canine cell/3873D (WRD) allows for robust viral replication and the production of progeny viruses [3, 4], the WRD cell line has become an ideal in vitro model for studying the pathogenesis of MVC. MVC, with its genetic tractability and reliable infectivity, is a suitable model for studying the interactions between the MVC capsid protein and the host, and for elucidating the infection mechanisms of bocaviruses.”

Here are the related references: 

  1. Blackburn, S.D.; Cline, S.E.; Hemming, J.P.; Johnson, F.B.; Attachment of bovine parvovirus to O-linked alpha 2,3 neuraminic acid on glycophorin A. Arch Virol 2005,150 (7), 1477-84.
  2. Sun, Y.; Chen, A.Y.; Cheng, F.; Guan, W.; Johnson, F.B.; Qiu, J.; Molecular characterization of infectious clones of the minute virus of canines reveals unique features of bocaviruses. J Virol 2009,83 (8), 3956-67
  3. 3. Binn, L.N.; Lazar, E.C.; Eddy, G.A.; Recovery and characterization of a minute virus of canines. Infect Immun 1970, 1 (5), 503-8.
  4. 4. Li, F.; Zhang, Q.; Yao, Q.; The DNA replication, virogenesis and infection of canine minute virus in non-permissive and permissive cells. Virus Res 2014, 179, 147-52.

Comment 2: Within the introduction there are a number of different sections, including one describing the MVC virus, another addressing the pathological effects of the virus on dogs and another introducing the WRD cell system and ROCK1 proteins, in which regard it would be necessary to merge the sections with more sentences between each section.

Response 2: Thank you for pointing this out. We have revised the introduction by adding more transitional sentences to better merge these sections and improve the overall coherence. 

Comment 3: It is requested that the authors be more precise with respect to the experimental conditions and methods used, e.g., with respect to transfection steps and cell cultures. What is meant by standard conditions?

Response 3: Thank you for pointing this out. We have provided more detailed descriptions of transfection steps and cell culture procedures.

Comment 4: The section on transfection is clear and correctly written. It would be necessary to write exactly how the treatments are chosen and how this may affect the experiment.

Response 4: Thank you for your positive feedback on the transfection section. This is a very good question. Our team has indeed conducted validation on treatment choices, but the experimental results are presented in another article by our team. However, that paper has not been published yet. To validate the aforementioned findings, a pretreatment was established where in WRD cells received RhoA inhibitor CCG-1423 (4 μM) or ROCK1 inhibitor Y27632 (20 μM) under controlled conditions (37°C, 5% COâ‚‚) for 1, 3, and 5 hours before MVC infection (15 min, MOI = 1). Western blot analysis showed that the expression of ROCK1 and RhoA was significantly downregulated compared to the untreated infection control. (Fig. A, B). Morphometric analysis of immunofluorescence images (40×) revealed preserved cellular architecture following 1 h pretreatment, whereas extended durations (3-5 hr) induced cytoskeletal reorganization, evidenced by reduction in cell spreading area (Fig. C). Based on these findings, 1 h pretreatment was selected for subsequent mechanistic studies. 

Fig. Effect of RhoA/ROCK Pathway Inhibitors on RhoA and ROCK1 Expression and Cell Morphology. WRD cells were pretreated with RhoA inhibitor CCG-1423 (4 μM) or ROCK1 inhibitor Y27632 (20 μM)  for 1 hour, 3 hours, and 5 hours, followed by MVC infection for 15 minutes. (A, B) A western blot was employed to evaluate the expression levels of RhoA and ROCK1, respectively. (C) Confocal microscopy analysis of the effects of inhibitors on cell morphology. Immunofluorescence analysis was performed using anti-Occludin (1:100) antibody, followed by incubation with AlexaFluor 488 conjugated secondary antibody (1:200). The scale bar represents 20 μm. * P < 0.05; ** P < 0.01; *** P < 0.001.

Comment 5: The discussion describes all the different aspects that have been studied. More details should be included with respect to the molecular mechanism of TJs disruption. Furthermore, some studies should be added concerning the modulation of the RhoA/ROCK pathway. A better definition of the future goal is requested.

Response 5: Thank you for your comprehensive suggestions. We have expanded the discussion by adding more details on the molecular mechanism of TJs disruption, citing relevant studies on the modulation of the RhoA/ROCK pathway, and provided a clearer definition of future goals, from line 559 to line 602 (Manuscript.pdf).

Reviewer 2 Report

Comments and Suggestions for Authors

The study enlightens the mechanism of virulence of the minute virus of canines via an in vitro experimentation accompanied by biochemical and molecular analyses. These analyses are totally comprehensive and the study is well written, whereas the final hypotheses investigated are supported and confirmed by the results. I would pleasantly recommend publication and I have some minor suggestions to improve the quality of the manuscript, as follows:

Firstly in the introduction a short presentation of data regarding the world prevalence, morbidity and mortality of the diseased animals would benefit the study.

In Materials and Methods, a better explanation of the the tenfold dilutions for the quantification is also needed.

In the results, please add if necessary, the Viability of WRD cells was always not significantly different?

Also in Figures 7 and 9, the levels of significance above the histogram within the graph are confusing. I suggest to put an asterisk above each bar instead of all together

Finally, to better present the outcomes, a graphical depiction of the proposed mechanism would also benefit the manuscript.

Author Response

Dear Reviewer,

Thank you for the editor and reviewers’ valuable comments on our manuscript entitled “RhoA/ROCK1/MLC2 Induced Positively by MVC Modulates Tight Junction Dissociation and Facilitates Occludin-Mediated Viral Infection” (Manuscript ID: microorganisms-3468333). We thoroughly revised the manuscript based on the comments.

Reviewer comments:

Comment 1: Firstly, in the introduction, a short presentation of data regarding the world prevalence, morbidity, and mortality of the diseased animals would benefit the study.

Response 1: Thank you for this valuable suggestion. We have briefly outlined the prevalence of MVC in the introduction. The incidence of canine bocavirus shows a significant regional variation. According to some clinical surveys, the detection rate of canine bocavirus is more commonly found in dogs with respiratory or gastrointestinal diseases. Studies indicate that some healthy dogs may also carry the virus without exhibiting obvious clinical symptoms. Overall, the incidence varies significantly among dog populations in different regions. Therefore, it is difficult to present specific data on incidence and mortality rates in the text. Here are the related references:

  1. Yan, N.; Yue, H.; Kan, R.; A novel recombinant genome of minute virus of canines in China. Arch Virol 2019, 164 (3), 861-865.
  2. Campalto, M.; Carrino, M.; Tassoni, L.; Divergent minute virus of canines strains identified in illegally imported puppies in Italy. Arch Virol 2020, 165 (12), 2945-2951.
  3. Weber, M.N.; Cibulski, S.P.; Olegario, J.C.; Characterization of dog serum virome from Northeastern Brazil. Virology 2018, 525, 192-199.

Comment 2: In Materials and Methods, a better explanation of the tenfold dilutions for the quantification is also needed.

Response 2: Thank you for pointing out this issue. We have provided a more detailed explanation of the tenfold dilutions for quantification in the Materials and Methods section.

Comment 3: In the results, please add if necessary, the Viability of WRD cells was always not significantly different?

Response 3: Thank you for this advice. Statistical analysis showed that for CCG-1423, differences were not statistically significant at concentrations below 3 µM. For Y27632, a phenomenon of low-dose inhibitor-induced promotion of cell viability was observed at concentrations below 20 µM. Based on literature references, the selected working concentration is 4 µM for CCG-1423 and 20 µM for Y27632.

Here are the related references:

 Sharif, M.; Baek, Y.B.; Naveed, A.; Stalin, N.; Kang, M.I.; Park, S.I.; Soliman, M.; Cho, K.O.; Porcine Sapovirus-Induced Tight Junction Dissociation via Activation of RhoA/ROCK/MLC Signaling Pathway. J Virol 2021, 95 (11).

Comment 4: Also, in Figures 7 and 9, the levels of significance above the histogram within the graph are confusing. I suggest putting an asterisk above each bar instead of all together

Response 4: Thank you for the clear suggestion. We have modified Figures 7 and 9 by placing an asterisk above each bar to avoid confusion regarding the levels of significance.

Comment 5: Finally, to better present the outcomes, a graphical depiction of the proposed mechanism would also benefit the manuscript.

Response 5: Thank you for the insightful proposal. We have already included a graphical representation in Figure 10D, illustrating the mechanism of RhoA/ROCK1/MLC2 signaling pathway activation following MVC infection.

Reviewer 3 Report

Comments and Suggestions for Authors

Comments to MVC manuscript:

This manuscript describes the interaction of the VP2 protein of the MVC (Minute Virus of Canines) virus and the kinase domain of RhoA-associated protein kinase 1 (ROCK1) expressed in the Walter Reed Canine cell line 3873D (WRD). They further show that the RhoA/ROCK1/myosin light chain 2 (MLC2) signaling pathway is activated upon MVC infection, and the authors propose that the activation of the RhoA/ROCK1/myosin light chain 2 (MLC2) signaling pathway leads to the exposure of the tight junction protein Occludin, which in turn increases viral infection since it functions as a receptor for the VP2 protein of the MVC. 

A major question is that in order to reach ROCK1 which is an intracellular protein, the virus first has to enter the cells, meaning that initially an interaction with a surface molecule is required before the virus VP2 protein can interact with ROCK1. The abstract needs to add some words about this sequalae.

The method section is incomplete and should describe in detail all procedures done.

Also, the result section needs rephrasing after scientific editing has been done. There are so many inaccuracies. The data are nice, but these should be described in a proper way. Please ensure that all WB are the right ones without duplication. Duplication has been detected visually by the reviewer, which makes some figures problematic. There are some very suspicious panels, especially of β-actin with the same smile pattern and similar overlaps between lanes, sometimes with different exposures. This has to be verified by all authors, and corrected accordingly.

Specific comments:

  • Line 19: Instead of "each inhibitor decreased" which has not been mentioned before, I would suggest writing: "Inhibition of either of these components decreased".
  • Line 55: Please mention which host cell surface receptors are involved in the initial viral adhesion.
  • Line 59: How was WRD transformed into an epithelial cell line?
  • Line 95: Please spell out the abbreviation MLC2.
  • Line 98: Please spell out HA.
  • The antibody catalog numbers should be added to the text.
  • Line 107: Please write the abbreviations in full name.
  • The sentence in line 107-109 should be rephrased as in its current state is misleading (it says that all eukaryotic vectors use… but it is only for this paper…). For instance, you can write: "SalI/XhoI restriction enzyme cleavage sites were used for the pCMV-HA vector, while the EcoRI/XhoI restriction enzyme cleavage sites were used for the pGEX-4T-1 vector.
  • Line 109: The following can be deleted: "Upon completion of synthesis," (you did cloning).
  • Table 1 should include definition of abbreviations (RBD, CRD).
  • Line 110: A reference for the pXJ40-Flag-VP2 plasmid should be provided. And you need to state that this is a eukaryote vector. The cell line using this vector should be mentioned, including growth conditions.
  • Line 112: The co-immunoprecipitation assay should be described here. The sentence of confirmed interaction belongs to Result section.
  • Line 116: Please correct to: coli. (Also, bacterial names should be in italics).
  • The amount of ampicillin in the agar plates should be stated.
  • Line 117: Please check the text. It says that the IPTG was added to the agar plates, but I believe that this was added to the LB broth medium after picking up bacteria from the ampicillin agar plates. The full name of IPTG should be written as well as its source.
  • Line 118: You need to describe how the bacterial lysis was done.
  • Line 119: The text is not logical. The bacteria are sonicated to get the lysate. When you have got the lysate, there is no need for sonication. Please correct the text.
  • The buffers used for GST affinity column should be stated. You need to mention that cloning in pGEX-4T-1 results in a GST-tagged protein, and to mention which negative control was used/
  • In line 121: The three steps should be described.
  • Line 122: BCA should be spelled out and the source mentioned.
  • Line 123: The composition of the Coomassie Brilliant Blue should be described.
  • Line 126: The amount of beads and the buffer should be mentioned. Also, the amount of bacterial lysate, and initial bacterial volume should be stated.
  • Lines 126-127: The sentence should be in the past tense.
  • The preparation of the cell lysate containing VP2 protein should be described. Including host cells, lysis buffer, and purification.
  • Line 129: The incubation buffer should be stated.
  • Line 130: You need to state how many times the beads were washed, and which washing buffer was used.
  • Line 131: The source and dilutions of the antibodies should be stated.
  • Line 133: The stock concentration should be stated.
  • Line 135: The cell density seeded should be mentioned together with the volume of the media.
  • Line 137: You cannot say "eliminated". I think you meant "removed". Please correct.
  • Line 137: The concentration of CCK-8 solution and the media used have to be mentioned.
  • Line 138: The CO2 level should be stated.
  • Line 139: The source of the ELISA reader should be mentioned.
  • Line 135: The number of WRD cells seeded prior to transfection should be stated. Also, the medium used during transfection should be mentioned.
  • Line 147: The name of the transfection reagent should be stated.
  • Line 149: Describe how did you know when the gene was silenced.
  • Line 152: You need to mention the incubation time with MCV. According to the sentence, the inhibitors were removed prior to MVC infection. Is this correct?
  • Line 163: You have to state which plasmid.
  • Line 165: The compositions of the RIPA buffer and the protease inhibitors should be described. Also, the amount of cells to volume should be stated.
  • Line 166: The kind of sonicator should be stated.
  • The source of the Protein A/Protein G beads should be stated.
  • Line 169: I think "incubated" is a better word than "treated". This is not a treatment, but an immunoprecipitation protocol.
  • Line 174: Again, you need to state the number of cells and volume of media. How do you determine the MOI, when cells are seeded the day before experiment?
  • Line 174: The source of " 6-well chamber slides" should be stated.
  • Line 177: Is not 5% Triton X-100 a really high concentration? Please check the concentration. Also, the solvent used should be stated.
  • Line 182: The dilution of the secondary antibody should be stated.
  • The concentration of DAPI and the solvent should be stated, as well as the source.
  • Line 186 – You need here to refer to Section 2.12.
  • Line 188: Please correct to "Qiagen". (In general, please double check all information in the method section).
  • Line 189 lacks cDNA preparation prior to qPCR.
  • Line 197: This is the only time that pIMVC appears in the text. You need to define what pIMVC is.
  • Section 2.2 You need to add what you detect in the title. Not sufficient to say "flow cytometry".
  • Line 212: Correct to "visualization"
  • Line 216: Please provide a figure for the concave structure of VP2.
  • Line 217: Add space before bracket.
  • Lines 220-222: The methods should be described in detail in the method section. Here you should simply describe that VP2 was immunoprecipitated to determine interacting proteins.
  • Lines 222-223: Again, this belongs to method section, and you should describe what you observed in the result section. Also, it is not the VP2 band that underwent MS analysis, but I anticipate the whole lane – as the MW of interacting proteins not necessarily migrate together with VP2 in the SDS-PAGE, but accordingly to their MW. It seems that the person that has written the manuscript has totally misunderstood the principle of the methodologies.
  • Line 223: You detected these proteins, but not screened them, so you need to correct the text. Please let a scientific expert read the revised manuscript to ensure correctness in the sentences.
  • Line 225: If you increased the stringency of co-IP, would fewer interacting proteins be detected? Would you better reach specificity? Could it be due to technical issues?
  • Line 226: I think you meant "categorized" and not "prioritized".
  • Line 228: A period should be added after the last parenthesis. A new sentence should start with "Three proteins".
  • Line 229: The functions of the three proteins should be described. Especially ZO2, and TMED10.
  • Line 239: Please say "interaction" instead of "connection". These are two quite different words with different meanings.
  • Line 232: Describe which effectors are known to be involved?
  • Line 232: Please write "focused" instead of "concentrated".
  • The raw data of Figure 1A should be provided in a Supplementary Table.
  • Line 236: The title should be the conclusion of the Figure, namely VP2 interacts with ROCK1.
  • Line 240: The reasoning of the assay should be in the main text. Here you should describe exactly what you did. (B): "and immunoprecipitated with anti-VP2 or control antibody followed by Westen blot (WB) using the indicated antibodies." Similar corrections should be done in C and D. The sentence that was written in the legend should be part of the main text.
  • Line 242: There is an extra point that should be deleted.
  • Line 246 and elsewhere. Please add space before the brackets.
  • Line 246: Correct to "reorganizes".
  • Line 248: Please correct to: "possesses".
  • Line 249: Please correct to: "which functional…. can interact…".
  • Line 252: I can cry when reading this "HA tag derived from rabbits". HA is an epitope (amino acids 98-106) of the influenza hemagglutinin (HA) protein. The antibody against HA is made in rabbit. So please correct the text. You need to state if the HA-tag is N- or C-terminal. And the animal species of ROCK1.
  • In all of the WB figures, the MW of the bands should be stated.
  • Line 254: You didn't use a "tagged" antibody, but an antibody recognizing the tagged protein (HA or Flag). Please be accurate in the text.
  • Line 254: What do you mean by "strong" – you detected an interaction. So "strong" can be deleted.
  • Lines 258: And if you transfected the cells with VP1, would you also see an interaction? What is the molecular weights of VP1 vs. VP2? – can these be distinguished on SDS-PAGE? You actually did ip with anti-HA or anti-Flag, so there should theoretically not be any interference from VP1.
  • Some of the inputs in Figure 2B are low. Can higher exposure time be shown for these?
  • Figure 2B, the panel of WB: Flag of VP2 has cut the lower band. VP2 appears as two bands in upper WB, while in one band in lower WB. It means that only one of them interacts with the kinase. This should be discussed in the text.
  • Line 262: Correct to: co-IP.
  • Line 266: You need to indicate in which subfigures you used control mouse or rabbit IgG, and which isoform.
  • Line 267: You need to mention the substrates of ROCK1.
  • Please state why you decided to clone the eukaryotic in a prokaryotic vector, and not in a eukaryotic vector.
  • Figure 3: You need to state in which subfigure you used which antibody for WB. I have a problem with Subfigure B in the last lane, how can it be that you have VP2 when you do not have ROCK1-KINASE-GST? The same for first line, if you did not use GST-beads, how could you get the pull-down of the protein? Why did you add GST to GST-ROCK1-kinase? Please clarify and make the corrections. Also, the input should be shown for all samples.
  • Line 283: You need to add reference and explain which dynamic changes occur.
  • Line 284: IFA should be spelled out.
  • Section 3.3: The nuclear translocation is interesting. Which of the proteins has an NLS?
  • Figure 4B you need to label as in A, what was each transfection. The reader can guess, but it must be shown in the figure. How does the VP2 transfection in COS-1 cells affect cell viability?
  • Describe which substates does ROCK1 use in the nucleus. Which signaling pathways does ROCK1 induce in the nucleus?
  • Line 312: TJ should be spelled out.
  • Line 319: What happens after 60 min? You cannot say fluctuation at 60 min, It is upregulated. Fluctuations is when it is going up and down. How many times have this experiment been repeated? and you got similar data?
  • Figure 5: mpi should be defined. The number of repeats should be stated in the legend. I have difficulty seeing the subtle increase at 15 min for total RhoA and ROCK1. Also, a spelling mistake in Figure 5A: The number 1 is written instead of 15. Please correct. What about phosphorylation of ROCK1? The increase in phosphorylation of MLC2 is clear, and it seems to be more than the 1.5-fold change as indicated in the bar graphs. The increase in total MLC2 was only subtle, and not so much as stated in the bars. Please remeasure using a better program than Image J. The p-MLC2/MCL2 ratio should also be measured. Why is it written p-MLC and not p-MLC2 in the figure? If A and B are the same samples, and same actin, they can be put together.
  • Line 328: You need here first to introduce the reader to what you are studying before stating results. It is a jump from the previous paragraph, and the reasoning behind the experiment should be described. Also, which two inhibitors you are speaking about should be stated.
  • Line 332 – you need to state the inhibitors, what they are inhibiting instead of "chemicals". Put attention that there should be a space after the parentheses.
  • Line 337: Why should RhoA inhibitor CCG-1423 reduce the RhoA level? It should prevent its activity. Please explain in text. The same question for the other inhibitor. How do the inhibitors reduce these levels?
  • Why should DMSO reduce p-MLC levels? The p-MLC/MLC2 ratio should be calculated.
  • The densitometric calculations in Figure 7 do not seem reasonable. Please recalculate with a reliable program.
  • Why is MLC2 downregulated with siROCK1? What was the viability of the cells when doing siRNA transfection? A siRNA control is needed. Is the expression of ROCK1 required for MLC2 expression?
  • Figure 7 legend states that Occludin is analyzed, but it does not appear in the figure.
  • Line 346: syntactic problem in sentence. Please correct. E.g., WRD cells were pretreated with…, and then…".
  • Line 350: Add the concentration of ROCK1 inhibitor used.
  • Line 254: Should be corrected. The statistical analysis is at the right side of the WB! Also, correct to: "are displayed".
  • Figure 7: NC should be defined in legend. Was the MLC2 in B done on the same blot as p-MLC? Or a different one. The small bubble is only seen in p-MLC. Why is the MLC2 in A with double band, while in B it is a single band? It differs from p-MLC with two really separate bands (which has been partly cut in B). In C the p-MLC appears only in one band. How can this be explained?
  • Why was Occludin not mentioned in line 229?
  • Figure 8: Spelling mistake in A and B. Correct to: Occludin
  • Please also correct the text in figure legend 8B. You show two figures in B. The lower is exactly the same as A (exactly same panel of WB occluding – just a slight stronger exposure – you can see the 4th lane with the same half-smile). WB: VP2 can be transferred to A. Panel C: Controls are lacking: anti-HA in cells transfected only with pFlag-VP2, and cells transfected only with pHA-occludin. Sometimes VP2 appears as one band, and sometimes with two bands. How do you explain this?
  • Figure 8A: Stronger exposures of Claudin and ZO-1 should be shown. Figure 8A: Why is the Occludin band after IP weaker than the input? I would have expected the opposite.
  • Figure 8C: the upper panel is not synchronized with the lower panel.
  • Figure 8: The MW should be added to all figures.
  • In the attached full blot images – the MW marker should be added.
  • Line 378: This seems to be a comment provided by the mentor….
  • Figure 9: The parallel expression of ROCK1, RhoA, ZO1 and claudin should be included in this experiment. DMSO alone had also an effect. What was the final concentration of DMSO and the concentration of the inhibitors?
  • Again, in figure legend 9, the text is as a suggestion of an experiment in imperative instead of describing what was done (e.g., Pretreat cells). Instead of writing the specified dose, write the dose. The letters in the graphs are too small to see it.
  • Figure 9: The effect of the inhibitors in the absence of MVC should also be shown.
  • Line 398: The extra point should be deleted.
  • Although in figure legend the authors suddenly state that the determination of MV genome copy number is described in the Method section, this cannot be found. Please add this section. This sentence (lines 431-433) should be removed from the legend.
  • Figure 10: Panel A and D should be put together, as B and C. How can it be that VP2 and NS1 can not be detected after 6 h infection? Confer Figure 4. A higher exposure of VP2 should be presented in E.
  • Why is a higher exposure time shown for NS1 in F than in E and G?
  • Please verify the β-actin in Figure 10G. It is too similar to another panel (5A), but with lower exposure.
  • Are the inhibitors specific?
  • Line 456: But you need already Occludin expression for the initial viral entry.
  • What would happen if you add both CCG-1423 and Y27632?
  • The labeling of the densitometry of E, F and G should be clearer. g., Control, CCG-1423 at the top and VP2 gene expression in Y-axis for the very left one, and such forth. In the current state it is misleading.
  • There is a need to do an overexpression of ROCK1 to see its effect on Occludin expression and virulence.
  • Line 442: Add reference to sentence starting with "Many studies".
  • Line 443: A syntactic error: correct to "change" instead of "changes in". (You wrote could).
  • Figure 11C: Flow cytometric analysis is not convincing. Autofluorescence histogram has to be added to each of them to see where to put the marker.
  • The magnification of Figure 11B is much higher than the magnification of Figure 11A, such that the size bar should be different. It would be preferable to show both this magnification of Fluo-3, but also a panoramic image as in A, to see the general trend.
  • Figure 11D lacks the link between the viral VP2 protein and ROCK1 activation. The TJ lines should also be defined in the figure as you did for myosin, virus and Occludin. The actin belt should be proven by immunofluorescence.

Comments on the Quality of English Language

English editing is required.

Round 2

Reviewer 3 Report

Comments and Suggestions for Authors

The title is quite long and cumbersome and should be rephrased to be clearer and the abbreviation MVC should be spelled out. You need to think about what you want to emphasize. I think the word "positively" is unnecessary. Maybe: "The Minute Virus of Canines (MVC) activates the RhoA/ROCK1/MLC2 signal transduction pathway resulting in the dissociation of tight junctions and interaction between the viral protein 2 (VP2) and occludin."

English editing is required. Especially of the abstract.

Line 12: VP2 should be spelled out, as this is the first time mentioned.

Line 23: Did you intend "prevented" (instead of "restored").

Line 25: Correct to "the two inhibitors".

Line 35. There is a syntactic problem in the sentence.

Line 177: The name of the transfection reagent should be stated.

Line 250: I think you intended μg/μL.

Comments on the Quality of English Language

English editing is required.
